# Node Classification With Reject Option

**Uday Bhaskar Kuchipudi**                                    *udaybhaskar.k@research.iiit.ac.in*
*Machine Learning Lab, IIIT Hyderabad, India*

**Jayadratha Gayen**                                    *jayadratha.gayen@research.iiit.ac.in*
*Machine Learning Lab, IIIT Hyderabad, India*

**Charu Sharma**                                    *charu.sharma@iiit.ac.in*
*Machine Learning Lab, IIIT Hyderabad, India*

**Naresh Manwani**                                    *naresh.manwani@iiit.ac.in*
*Machine Learning Lab, IIIT Hyderabad, India*

**Reviewed on OpenReview:** *https://openreview.net/pdf?id=4xXJDO8Bvu*

## Abstract

One of the key tasks in graph learning is node classification. While Graph neural networks have been used for various applications, their adaptivity to reject option settings has not been previously explored. In this paper, we propose NCwR, a novel approach to node classification in Graph Neural Networks (GNNs) with an integrated reject option. This allows the model to abstain from making predictions for samples with high uncertainty. We propose cost-based and coverage-based methods for classification with abstention in node classification settings using GNNs. We perform experiments using our method on standard citation network datasets Cora, CiteSeer, PubMed and ogbn-arxiv. We also model the Legal judgment prediction problem on the ILDC dataset as a node classification problem, where nodes represent legal cases and edges represent citations. We further interpret the model by analyzing the cases in which it abstains from predicting and visualizing which part of the input features influenced this decision.

## 1 Introduction

In recent times, we have witnessed a surge of interest in GNNs and their applications in various domains such as computer vision (Satorras & Estrach, 2018), natural language processing (Schlichtkrull et al., 2017), and bioinformatics (Xia & Ku, 2021), to name a few. GNNs (Kipf & Welling, 2017) capture structural aspects of the data in the form of nodes and edges to perform any prediction task. It learns node, edge, and graph-level embeddings to get high-dimensional features using the message-passing mechanism in the GNN layer. GNNs are also used in high-risk applications such as legal judgment prediction (Dong & Niu, 2021), disease prediction (Sun et al., 2021), financial fraud prediction (Xu et al., 2021), etc., with a high cost of incorrect predictions. In such high-risk situations, standard GNN models are ineffective in handling uncertainty.

Uncertainty estimation approaches measure the prediction uncertainty involved in high-risk applications. Conformal prediction methods are popular for uncertainty estimation (Gawlikowski et al., 2023; Wang et al., 2024; Angelopoulos & Bates, 2021).

Another approach to handling the uncertainty in high-risk scenarios is using a reject option in the classifier. The objective is to avoid making any decisions on difficult and confusing examples. Consider the case of the diagnosis of a patient for a specific disease. In case of confusion, the physician might choose not to risk misdiagnosing the patient. She might instead recommend further medical tests to the patient or refer her to

an appropriate specialist. The primary response in these cases is to "reject" the example. Such flexibility of the classifier to avoid taking any decision is called the reject option.

Reject option classifiers have been extensively used in various high-risk applications such as healthcare (Hanczar & Dougherty, 2008; da Rocha Neto et al., 2011), finance (Rosowsky & Smith, 2013) etc. Approaches for learning reject option classifiers can be divided into two broad classes: (a) cost-based and (b) coverage-based. In cost-based approaches Kalra et al. (2021); Charoenphakdee et al. (2021); Ramaswamy et al. (2018); Cao et al. (2022), the goal of the algorithm is to find an optimal classifier by minimizing a loss which also incorporates the cost of rejection along with the cost of misclassification. On the other hand, in the coverage-based method Geifman & El-Yaniv (2019; 2017), a coverage parameter is pre-specified, and the algorithm tries to maintain the fraction of unrejected samples the same as coverage. Both categories try to reject difficult examples.

In this paper, we propose integrating a reject option in GNNs for the node classification problem. We propose two variants corresponding to cost-based and coverage-based approaches. We also present how this method can be used in real-world scenarios by working on prediction tasks of high-risk domains such as Healthcare and Law. Our contributions in this paper are as follows: *i*) We extend and generalize GNNs to train for node features with cost-based and coverage-based abstention models. *ii*) We perform an empirical study to evaluate our models on popular benchmark datasets for node classification tasks and compare them with baseline methods. *iii*) We show extensive results of our method on the Indian Legal Documents Corpus (ILDC) dataset for the LJP task. *iv*) To understand why our model chooses to reject certain cases, we further examine these cases with the help of Shapley Additive Explanations (SHAP) (Lundberg & Lee, 2017).

## 2 Related Work

### 2.1 Node Classification

Node classification is a fundamental task related to machine learning for graphs and network analysis. GNN methods can be broadly classified into three categories that perform node classification as the primary task. The first set of models introduced convolution-based GNN architectures by extending original CNNs to graphs (Scarselli et al., 2008; Defferrard et al., 2016; Hamilton et al., 2017; Kipf & Welling, 2017; Bresson & Laurent, 2017). Secondly, proposed attention and gating mechanism-based architectures using anisotropic operations on graphs (Veličković et al., 2018). The third category focuses on the theoretical limitations of previous types (Xu et al., 2018; Morris et al., 2019; Maron et al., 2019; Chen et al., 2019).

### 2.2 Reject Option Classification

There are two broad categories of approaches for reject option classifiers: coverage-based and cost-based. Coverage is defined as the ratio of samples that the model does not reject. For a given coverage, the model finds the best examples that can give the best performance. SelectiveNet is a coverage-based method proposed for learning with abstention (El-Yaniv et al., 2010; Geifman & El-Yaniv, 2019). SelectiveNet is a deep neural network architecture that optimizes prediction and selection functions to model a selective predictor. As this approach does not consider rejection cost $d$ in their objective function, it can avoid rejecting hazardous examples.

Cost-based approaches assume that the reject option involves a cost of $d$. The cost of rejection is much smaller compared to misclassification. Overall, these approaches aim to minimize the number of rejected examples as well as minimize the misclassification of unrejected samples. Kalra et al. (2021) propose a deep neural network-based reject option classifier for two classes that learn instance-dependent rejection functions. Ramaswamy et al. (2018) multiclass extensions of the hinge-loss with a confidence threshold are considered for reject option classification. Ni et al. (2019) prove calibration results for various confidence-based smooth losses for multiclass reject option classification. Charoenphakdee et al. (2021) prove that $K$-class reject option classification can be broken down into $K$ binary cost-sensitive classification problems. They subsequently propose a family of surrogates, the ensembles of arbitrary binary classification losses.

Cao et al. (2022) propose a general recipe to convert any multiclass loss function to accommodate the reject option, calibrated to loss $l_{0d1}$. They treat rejection as another class at the time of prediction.

## 2.3 Uncertainty Estimation Using Conformal Prediction

Uncertainty in Deep Neural Networks (Gawlikowski et al., 2023) studies the sources of uncertainty, like data uncertainty and model uncertainty, and the estimation of uncertainty measures to be further used in high-risk applications. This is further explored in the Graph setting by Wang et al. (2024), exploring both sources of uncertainty and estimating uncertainty. In the GNN setting, uncertainty measures are further utilized for downstream tasks like OOD detection, outlier identification and Trustworthy GNNs. Conformal Prediction (Angelopoulos & Bates, 2021) uses these uncertainty measures to predict an interval instead of one class, which will have the true label with a fixed probability with statistical guarantees. This setting uses distribution free uncertainty quantification, which makes it accessible to a wide range of applications. Huang et al. (2023) propose conformal GNN (CF-GNN), extending conformal prediction to GNNs and node classification.

# 3 Method

GNN models like GAT (Veličković et al., 2018) focus on learning effective and efficient representations of nodes to perform any downstream task. Let $\mathcal{X}$ be the instance space and $\mathcal{Y} \in \{1, \ldots, K\}$ be the label space. We represent the embedding space learned using GNN by $\mathcal{H}$. GNN treats each instance as a node and learns an embedding for each node. We used GAT as the base GNN architecture, but our method is model agnostic and can be replaced with any GNN architecture to learn the node embeddings.

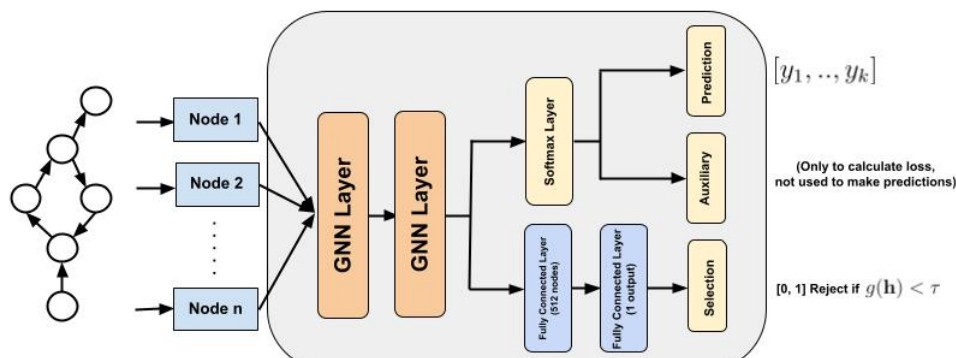

Figure 1: Architecture of NodeCwR-Cov: Coverage based node classifier with rejection.

## 3.1 NCwR-Cov: Coverage Based Node Classifier With Rejection

NCwR-Cov uses coverage-based logic to learn node classifiers with a reject option. We use similar ideas to SelectiveNet (Geifman & El-Yaniv, 2019) to learn the coverage-based rejection function. Figure 1 shows the architecture of NCwR-Cov. Node representations are learned using the first GNN layer and given as input to the second GNN layer which follows the softmax layer. The second GNN layer and softmax layer combined learn mapping $\mathbf{f} : \mathcal{H} \to \Delta_{K-1}$ where $\Delta_{K-1}$ is $K$-dimensional simplex. Function $\mathbf{f}$ is used to predict the class of a node. There are two more fully connected layers after the softmax layer (having 512 nodes and one node) to model the selection function $g : \mathcal{H} \to \{0, 1\}$. Selection function $g$ decides whether to predict a given example or not. Selection function $g(\mathbf{h})$ is a single neuron with a sigmoid activation. At the beginning, a threshold of 0.5 is set for the selection function, which means $\mathbf{f}(\mathbf{h})$ predicts $\mathbf{h}$ if and only if $g(\mathbf{h}) \geq 0.5$. The auxiliary prediction head implements the prediction task $a(\mathbf{h})$ without the need for coverage to get a better representation of examples with low confidence scores, which are usually ignored by the prediction head. This head is only used for training purposes. We use cross-entropy loss $l_{ce}$ to capture the error made by the prediction function $\mathbf{f}(\mathbf{h})$. The empirical risk of the model is captured as follows.

$$r(\mathbf{f}, g|S_n) = \frac{\frac{1}{n}\sum_{i=1}^{n} l(f(\mathbf{h}_i), y_i)g(\mathbf{h}_i)}{\phi(g|S_n)}$$

where $\phi(g|S_n)$ is empirical coverage computed as $\phi(g|S_n) = \frac{1}{n}\sum_{i=1}^{n} g(\mathbf{h}_i)$. An optimal selective model could be trained by optimizing the selective risk given constant coverage. We use the following error function to optimize $\mathbf{f}(.)$ and $g(.)$.

$$E(\mathbf{f}, g) = r(\mathbf{f}, g|S_n) + \lambda\Psi(c - \phi(g|S_n))$$

where $\Psi(a) = \max(0, a)^2$ is a quadratic penalty function, $c$ is the target coverage, and $\lambda$ controls the importance of coverage constraint. The loss function used at the auxiliary head is standard cross-entropy loss $l_{ce}$ without any coverage constraint. Thus, the empirical risk function corresponding to the auxiliary head is $E(\mathbf{f}) = 1/n\sum_{i=1}^{n} l_{ce}(\mathbf{f}(\mathbf{h}_i), y_i)$. The final error function of NCwR-Cov is a convex combination of $E(f, g)$ and $E(\mathbf{f})$ as follows, $E = \alpha E(\mathbf{f}, g) + (1 - \alpha)E(\mathbf{f})$, where $\alpha \in (0, 1)$. When the data is trained over a training set using a coverage constraint, this constraint is violated on the test set. The constraint requires the true coverage $\phi(g)$ to be larger than the given coverage constraint $c$, which is usually violated. To get the optimal actual coverage, we calibrate the threshold $\tau$ to select the example in $g(h')$ using this validation set, which results in coverage as close as possible to the target coverage.

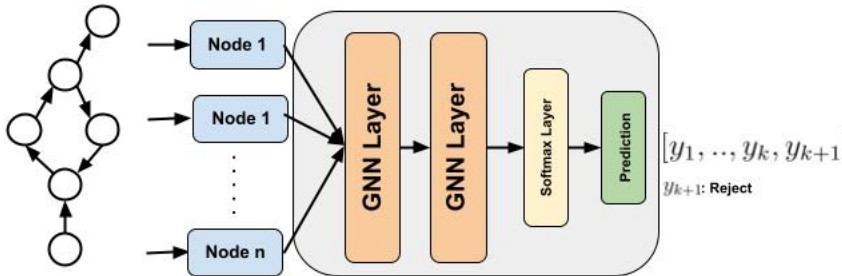

Figure 2: Architecture of NodeCwR-Cost: Cost based node classifier with rejection.

## 3.2 NCwR-Cost: Cost Based Node Classifier With Rejection

In the cost-based method, the cost of rejection $d$ is pre-specified. The goal here is to learn an optimal node classifier with rejection for a given $d$ value. The architecture of NCwR-Cost is presented in Figure 2. The first block in NCwR-Cost consists of two GNN layers. The output of the second GNN layer is fed to a softmax layer with $(K + 1)$ nodes. Note that we assume rejection as the $(K + 1)^{th}$ class here. The second GNN layer and softmax layer combined learn prediction function $f : \mathcal{H} \rightarrow \Delta_K$ where $\Delta_K$ is $(K + 1)$-dimensional simplex. Note that $(K + 1)^{th}$ output corresponds to the reject option in this architecture. Let $\mathbf{e}_j$ denote $K + 1$-dimensional vector such that its $j^{th}$ element is one and other elements are zero. Note that $(K + 1)^{th}$ element never becomes one as we do not get a rejection label in the training data. We use the following variant of cross-entropy loss, which also incorporates the cost of rejection (Cao et al., 2022).

$$l_{ce}^d(f(\mathbf{h}), \mathbf{e}_y) = l_{ce}(f(\mathbf{h}), \mathbf{e}_y) + (1 - d)l_{ce}(f(\mathbf{h}), \mathbf{e}_{K+1}) = -\log f_y(\mathbf{h}) - (1 - d)\log f_{K+1}(\mathbf{h}) \tag{1}$$

Here $f_{K+1}(\mathbf{h})$ is the output corresponding to the reject option, and $f_y(\mathbf{h})$ is the output related to the actual class. For very small values of $d$, the model focuses more on maximizing $f_{K+1}(\mathbf{h})$ to prefer rejection over misclassification. Note that loss $l_{ce}^d$ is shown to be consistent with the $l_{0d1}$ loss (Cao et al., 2022). For $d = 1$, the loss $l_{ce}^d$ becomes the same as standard cross entropy loss $l_{ce}$.

## 4 Experimental Setup

In this section, we provide the details of the experimental setup.

### 4.1 Datatsets Used

We evaluate our model on three standard citation network datasets, Cora, CiteSeer, PubMed (Sen et al., 2008) and an OGB Dataset ogbn-arxiv Hu et al. (2020). In these datasets, each document is represented by a node, and the class label represents the category of the document. Citations are represented by undirected edges. We follow standard practices (Kipf & Welling, 2017; Veličković et al., 2018) for training node classifier. We use 20 nodes per class for training, 500 nodes for validation and 1000 for testing on the three Planetoid datasets and the given splits in OGB dataset.

### 4.2 Base Graph Neural Network Architecture Used

We can use any GNN as the base architecture for both the methods. In our experiment section, we use GAT as the base architecture due to its effectiveness and popularity. We also note that in our experiments, changing the base GNN Architectures and their parameters, like the number of layers, did not affect the results by a lot. We present this analysis in Appendix B.

For the GAT architecture, we closely follow the experimental setup mentioned in Veličković et al. (2018). We modify the open-source GAT implementation by Antognini (2021) for our approach. We first apply dropout (Srivastava et al., 2014) on node features with $p = 0.6$. These node features, along with the adjacency matrix, are passed through a GAT Layer with 8 attention heads, where each head produces eight features per node. We use LeakyReLU as the activation function inside the GAT Layer with $\alpha = 0.2$. These outputs are concatenated for the first layer (64 features per node). Another dropout layer with the same probability follows this. This is passed through the final GAT layer with a single attention head, which takes 64 features per node and outputs $k$ features per node, where $k$ is the number of classes. It is passed to ELU (Clevert et al., 2015) activation function. The network output is passed through a softmax layer.

### 4.3 Details of NCwR-Cov Implementation

We use the GAT architecture to integrate the coverage-based reject option into the model as mentioned in Geifman & El-Yaniv (2019). The output of the final layer is passed through softmax for both prediction head $f$ and auxiliary head $h$. It is also passed through a hidden layer with 512 nodes, batch normalization (Ioffe & Szegedy, 2015), ReLU, an output layer with one node, and sigmoid activation to get a selection score $[0, 1]$.

The Prediction head and Selection head are concatenated together, and the selective loss is performed on this output. We set $\lambda = 32$ as the constraint on coverage to calculate this loss. Cross-Entropy Loss is performed on the output of the Auxiliary head but is not used for making predictions. A convex combination of these two loss values with $\alpha^l = 0.5$ is used for backpropagation. Once the model is trained, the coverage on the test data set when $\tau = 0.5$ will vary. However, since we have the selection scores of each node in the validation set, we sort them and select a $\tau$ value that matches the expected coverage.

### 4.4 Details of NCwR-Cost Implementation

We trained NCwR-Cost by changing the output of the GAT network with an extra class in the output layer. For a $k$ class classification problem, we change the model architecture to have $k + 1$ outputs and backpropagate using the CwR Loss (see eq.(1).

### 4.5 Baselines Used

To the best of our knowledge, we are the first to use reject option classifiers for node classification on high-risk applications. This makes it tough to compare with existing baselines to show the importance of our contribution. However, we introduce some changes in existing uncertainty estimation methods to model them as Reject Option Classifiers and use them as baselines.

- Softmax Response (SR): We treat the Softmax scores of the Vanilla GNN as an uncertainty measure and reject examples when the predicted class score is lower than some threshold. As we change the

thresholds, the rejection rate changes. We used the following threshold values to get different reject option classifiers using Softmax-Response: $[0.5, 0.6, 0.7, 0.8, 0.9]$.

- CF-GNN Huang et al. (2023): Conformal classifier with coverage parameter $\alpha$ predicts a label set such that the probability of actual label being present in the predicted label set is greater than or equal to $1 - \alpha$. We model the state-of-the-art Conformal Prediction method for GNNs as Reject Option Classifiers in the following way. For a given value of $\alpha$, we simply reject those examples for which the CF-GNN predicts more than one class label. Rejecting such examples makes sense because of the label ambiguity and it aligns with the basic principles used for rejection. Note that as we increase the coverage parameter $\alpha$ in the CF-GNN, the predicted label set size will reduce. For a higher value of $\alpha$, the CF-GNN predicts smaller label sets. Thus, the rejection rate should decrease as we increase $\alpha$. Different $\alpha$ values used are: $[0.1, 0.125, 0.15, 0.175, 0.2]$.

For both baselines, we used the same GAT architecture that we used for NCwR-Cost and NCwR-Cov.

## 5 Experimental Results

Here, we present experimental results for proposed approaches NCwR-Cost and NCwR-Cov. We also compare them with different baselines to highlight the importance of the proposed approaches. We repeat each experiment 10 times with random initialization and report the average accuracies.

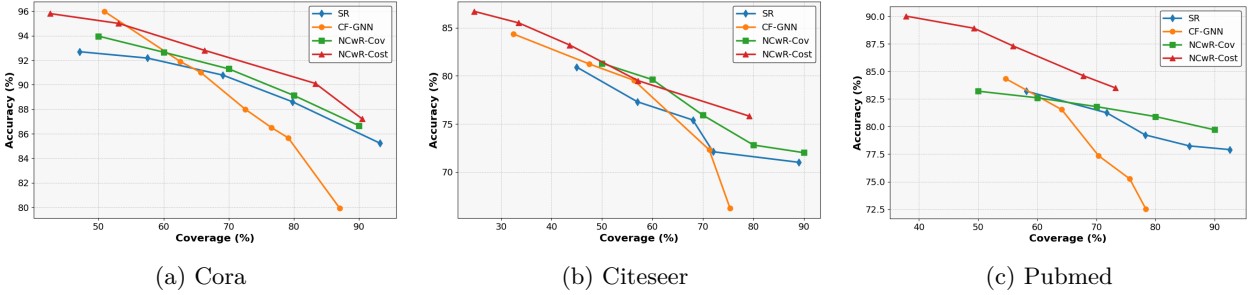

|         (a) Cora          |         (b) Citeseer          |         (c) Pubmed          |

Figure 3: Comparison of NCwR-Cost and NCwR-Cov with baselines.

### 5.1 Comparison Results with Baselines

Figure 3 shows comparison results with baselines. We observe that NCwR-Cost always outperforms the Softmax-Response based approach in terms of accuracy on unrejected samples for all coverage values. NCwR-Cost also outperforms the CF-GNN based approach for all coverage values and for all datasets except for one case in the Cora dataset. For the Cora dataset, for coverage of 50%, CF-GNN has marginally better accuracy. Thus, NCwR-Cost is a superior model compared to the baseline models.

We see that NCwR-Cov also outperforms the Softmax-Response based approach for all datasets and coverage values except for one case with Pubmed dataset (coverage value 60%). For coverage value 60% with the Pubmed dataset, Softmax-Response has slightly better accuracy than NCwR-Cov. Compared to CF-GNN, NCwR-Cov always performs better for coverage values greater than 60%. For coverage values smaller than 60%, CF-GNN performs marginally better than NCwR-Cov.

### 5.2 Results on NCwR-Cov and NCwR-Cost

**NCwR-Cov:** In Table 1, we report the performance of NCwR-Cov models trained for coverage rates ranging in $[0.5, \ldots, 0.9]$. Although we can calibrate the threshold to cover any number of examples irrespective of the training coverage, it is preferred to train the model on the same coverage rates and then calibrate $\tau$ to the same to get the best results. We observe that for all the datasets, as we increase the coverage, the accuracy on unrejected samples decreases. This pattern is expected as we reject lesser examples, there will be more difficult examples to classify.

| Coverage | Cora | CiteSeer | PubMed | ogbn-arxiv |
|---|---|---|---|---|
| 0.5 | $93.96 \pm 1.45$ | $81.30 \pm 2.19$ | $83.20 \pm 3.34$ | $79.46 \pm 2.16$ |
| 0.6 | $92.65 \pm 0.50$ | $79.60 \pm 2.43$ | $82.60 \pm 1.48$ | $77.92 \pm 1.73$ |
| 0.7 | $91.29 \pm 0.45$ | $75.90 \pm 2.86$ | $79.80 \pm 2.46$ | $75.21 \pm 1.41$ |
| 0.8 | $89.12 \pm 0.80$ | $72.80 \pm 1.05$ | $80.90 \pm 1.24$ | $73.12 \pm 2.25$ |
| 0.9 | $86.65 \pm 0.70$ | $72.00 \pm 0.69$ | $79.70 \pm 0.59$ | $72.1 \pm 0.63$ |
| 1.0 | 81.65 | 70.12 | 76.70 | 69.28 |

Table 1: Accuracy of NCwR-Cov for various coverage rates.

**NCwR-Cost:** In Table 2, we report the performance of NCwR-Cost models trained for rejection cost ($d$) taking values in $[0.5, 0.6, 0.7, 0.8, 0.85]$. As the cost of rejection $d$ increases, the rejection rate decreases. Decreasing the rejection rate will increase coverage. As the coverage increases, the model will misclassify more samples, which decreases the performance of unrejected samples.

| $d$ | Cora | | | CiteSeer | | | PubMed | | | ogbn-arxiv | | |
|---|---|---|---|---|---|---|---|---|---|---|---|---|
| | Acc | Cov | 0-d-1 | Acc | Cov | 0-d-1 | Acc | Cov | 0-d-1 | Acc | Cov | 0-d-1 |
| 0.5 | 95.8 ($\pm 0.05$) | 42.6 ($\pm 0.02$) | 0.305 ($\pm 0.11$) | 91.6 ($\pm 0.12$) | 9.7 ($\pm 0.05$) | 0.460 ($\pm 0.04$) | 88.9 ($\pm 0.02$) | 49.3 ($\pm 0.08$) | 0.309 ($\pm 0.16$) | 81.36 ($\pm 0.72$) | 40.29 ($\pm 0.15$) | 0.693 ($\pm 0.25$) |
| 0.6 | 95.0 ($\pm 0.04$) | 53.1 ($\pm 0.03$) | 0.308 ($\pm 0.06$) | 87.9 ($\pm 0.09$) | 17.6 ($\pm 0.09$) | 0.516 ($\pm 0.06$) | 84.6 ($\pm 0.05$) | 67.8 ($\pm 0.05$) | 0.298 ($\pm 0.08$) | 78.02 ($\pm 0.52$) | 58.75 ($\pm 0.01$) | 0.57 ($\pm 0.36$) |
| 0.7 | 92.8 ($\pm 0.04$) | 66.3 ($\pm 0.08$) | 0.283 ($\pm 0.07$) | 85.5 ($\pm 0.01$) | 33.5 ($\pm 0.02$) | 0.514 ($\pm 0.04$) | – | – | – | 75.15 ($\pm 0.41$) | 76.76 ($\pm 0.5$) | 0.446 ($\pm 0.62$) |
| 0.8 | 90.1 ($\pm 0.08$) | 83.3 ($\pm 0.04$) | 0.216 ($\pm 0.08$) | 79.5 ($\pm 0.11$) | 57.1 ($\pm 0.09$) | 0.460 ($\pm 0.02$) | – | – | – | 72.70 ($\pm 0.62$) | 94.15 ($\pm 0.13$) | 0.322 ($\pm 0.51$) |
| 0.85 | 87.2 ($\pm 0.06$) | 90.5 ($\pm 0.07$) | 0.196 ($\pm 0.06$) | 75.8 ($\pm 1.61$) | 79.2 ($\pm 0.04$) | 0.368 ($\pm 0.15$) | – | – | – | 70.63 ($\pm 0.34$) | 98.28 ($\pm 0.24$) | 0.308 ($\pm 0.11$) |
| 1 | 81.65 | 100.0 | 0.178 | 70.12 | 100.0 | 0.299 | 76.7 | 100.0 | 0.233 | 69.28 | 100.0 | 0.307 |

Table 2: Accuracy of NCwR-Cost for various cost-of-rejection values. We only present results on PubMed with $d \in [0.5, 0.6]$ as it has $k = 3$ classes and $d < \frac{k-1}{k}$.

**Comparison:** Figure 3 shows the coverage versus accuracy plots for both cost-based and coverage-based approaches on different datasets. The cost-based approach shows a clear advantage in terms of accuracy for most coverage rates. The reason is as follows. The coverage constraint in NCwR-Cov does not ensure the rejection of those examples that are hard to classify correctly. Thus, it may reject some of the easy examples. Thus, every coverage value may include more hard examples. We also observe a very high standard deviation in the performance of NCwR-Cov. On the other hand, NCwR-Cost prefers to reject hard examples first by assigning a cost to rejection. This makes NCwR-Cost perform better than NCwR-Cov.

### 5.3 Node Embedding Visualization

We plot t-SNE plots to represent the predicted class of each node. It is noticeable in Figure 4 that in both NCwR-Cost and NCwR-Cov models, the rejected examples (represented using black color) are usually the nodes that highly overlap between two or more classes. We can also notice that as the model coverage decreases, the number of examples it rejects increases and covers more overlapping boundaries between classes. It is worth noting that although the coverage and accuracy are almost comparable in both models, the examples that each model chooses to reject are from different overlapping classes. NCwR-Cost tries to reject those examples which are in the overlapping regions of different classes. On the other hand, NCwR-

Cov sometimes rejects examples which are predominantly from particular classes. For example, for coverage of 50.4%, NCwR-Cov rejects more examples of classes 3 and 4.

Figure 4: t-SNE plots representing predictions on Cora dataset (black - reject option).

# 6 Application: Legal Judgment Prediction

Legal judgment prediction (Feng et al., 2022; Cui et al., 2023) is an active area of research in the field of Machine Learning and Natural Language Processing. Automating the Judgment Prediction Process can be of huge value for various reasons. While these models already perform extremely well due to the recent surge in NLP progress, one issue that still remains is the reliability of these models to push to real-world scenarios. Unlike many current fields where NLP models are automating tasks, Legal Judgment Prediction is a very high-risk application in which the cost of misclassification is very high. In such high-risk applications, performing reliably well for a small set of examples is much more valuable than giving a prediction for every sample.

| | NCwR-Cost | | NCwR-Cov | |
|---|---|---|---|---|
| $d$ | Acc (%) | Cov (%) | Acc (%) | Cov |
| 0.25 | $87.24 \pm 2.45$ | $67.00 \pm 3.30$ | $97.55 \pm 0.62$ | 0.5 |
| 0.35 | $82.32 \pm 2.72$ | $86.34 \pm 4.61$ | $94.94 \pm 1.20$ | 0.6 |
| 0.40 | $79.95 \pm 1.52$ | $93.44 \pm 7.17$ | $90.58 \pm 1.24$ | 0.7 |
| 0.45 | $80.38 \pm 1.74$ | $97.83 \pm 0.76$ | $86.01 \pm 1.09$ | 0.8 |
| 0.50 | $79.94 \pm 2.09$ | $98.99 \pm 0.20$ | $81.87 \pm 1.03$ | 0.9 |

Table 3: Accuracy of NCwR-Cost and NCwR-Cov on ILDC dataset.

**Indian Legal Documents Corpus (ILDC)** (Malik et al., 2021) Indian Legal Documents Corpus is a dataset of a collection of case proceedings in English from the Supreme Court of India (SCI), covering the period from 1947 to April 2020 presented by Malik et al. (2021). The raw dataset poses significant pre-processing challenges due to unstructured document formats, spelling errors, and the need to remove meta-information and direct decision statements from the texts. The ILDC dataset is divided into two subsets: ILDC$_{single}$ - single petition cases, and ILDC$_{multi}$ - cases with multiple petitions leading to different

decisions. We worked with the ILDC$_{single}$ subset for our experimental setting. This subset contains a total of 7,593 cases, split into train/test/development sets (5,082/1,517/994).

This dataset was expanded by adding another $24,907$ cases without a final verdict and the citations within these cases with existing cases using ikanoon API by Khatri et al. (2023). This paper presented Legal Judgment Prediction as a semi-supervised node classification task where each node represents the text of the case proceedings, and a link represents a citation between cases. The additional cases without a label essentially enable the model with message passing through citation networks and are not part of the training set of nodes.

## 6.1 Experimental Setup

We follow the experimental setup presented in Khatri et al. (2023) and use a pretrained XLNet model from Malik et al. (2021) to extract language embeddings from all the new cases in the dataset. We formulate a graph where each node represents a case (Cases part of the ILDC have a label, and additional cases extracted by Khatri et al. (2023) do not have a label) and the citations between the cases as links. Khatri et al. (2023) presents that using directed or undirected edges does not affect the model performance by a lot; hence, we formulate the citations as undirected edges. On this graph, we train the GAT model for node classification and replicate the result presented in Khatri et al. (2023). On top of this, we perform experiments on this graph using our architectures. In Table 3 we report the performance for different cost of rejection values and coverage rates on the ILDC dataset.

## 6.2 Explainability

Explainability methods in machine learning aim to make model predictions understandable to humans, especially in high-stakes domains like law. SHAP (Shapley Additive Explanations) (Lundberg & Lee, 2017) is one of the most reliable and widely used approaches for model interpretability, based on cooperative game theory and Shapley values. In this work, SHAP is applied to explain predictions for legal judgment tasks, where understanding which parts of the legal text drive a model's decision is very useful for transparency. SHAP provides visual explanations by highlighting portions of the legal text. Specifically, red highlights indicate text that pushes the model toward a positive outcome (e.g., the model is more confident in its classification), while blue highlights signify text that supports a negative outcome (e.g., text that contradicts the model's classification). The intensity of these colours represents the magnitude of the contribution—the darker the shade, the stronger the influence of the text segment on the final prediction.

In our experiments, we use the last 512 tokens of a petition in the Supreme Court of India (SCI) Proceedings between the appellant and respondent, where the 'label' contains either '0' or '1'. A label of '0' represents petitions that have been rejected, while a label of '1' represents petitions that have been accepted.We have demonstrated two examples below: one where our model is very confident in its prediction and predicts correctly in Figure 5, and another where the model's confidence is lower, leading to an incorrect prediction in Figure 6.

### 6.2.1 Case 1: Explanation of the SCI Proceedings Where Model is Highly Confident

This case involves an appeal by A2, who was convicted under Section 380 of the Indian Penal Code (IPC) by the Trial Court for the theft of a gold chain. Two co-accused, A1 and A3, were acquitted of all charges. The appellant (A2) challenged the conviction in the Sessions Court and High Court, both of which upheld the conviction. The appellant then approached the Supreme Court, contesting the credibility of the prosecution's evidence, including the recovery of the stolen gold chain and the delay in filing the First Information Report (FIR).

The Supreme Court examined the evidence and found that the prosecution's case lacked credibility, particularly due to the significant delay in lodging the FIR (16 days) and the questionable recovery of the chain. As a result, the Supreme Court acquitted the appellant.

For this example, our model is very confident in its prediction and predicts the label '1' correctly. Explanation from SHAP is given in Figure 5.

Figure 5: SHAP explanation of Case where model prediction is right and confidence is high.

Figure 6: SHAP explanation of Case where model prediction is wrong and confidence is low.

**Sentences Leading to the Decision in Red**

- **Sentence 1** *"It is inconceivable that she would not realize that she had ..."*
  Highlights the Court's scepticism regarding the plausibility of the theft.

- **Sentence 2** *"As we have already noted that FIR was registered after about ..."*
  The Supreme Court found the delay in filing the FIR highly suspicious.

- **Sentence 3** *"The only evidence against him is the alleged recovery of the gold chain ..."*
  The Supreme Court questioned the reliability of the sole evidence used to convict the appellant.

**Sentence Contradicting the Decision in blue**

- **Sentence 3** *"The Trial Court further held ..."*
  Indicates that the recovery of the chain was crucial in linking the appellant to the crime.

### 6.2.2 Case 2: Explanation of the SCI Proceedings Where Model is very Low Confident

The case involves a dispute regarding the condition of certain boxes of goods. Mr. Gupta discovered that a few boxes were damaged, while the rest were intact and marketable. Mr. Banerjee, who was responsible for handling the boxes, claimed to have opened and repacked all 20 boxes, but his evidence was found unreliable. He admitted that he only opened a few boxes and fabricated his report on the condition of all 20 boxes. The Labour Court had previously concluded that the discharge order against Mr. Banerjee was unjustified. The Supreme Court reviewed the evidence and found Mr. Banerjee's actions to be questionable. The Supreme Court's decision favoured the appellant, overturning the Labour Court's conclusion that the discharge order was unjustified.

For this example, our model's confidence is lower, leading to rejection. Explanations from SHAP for this example are given in Figure 6, where the explanation is based on a wrong prediction. Below is the explanation based on the actual label. Thus, red and blue coloured explanations are reversed in Figure 6 with respect to the actual label.

**Sentences Leading to the Decision in Blue**

- **Sentence 1** *It is clear that the strapping is done . . ."*
  Highlights the unreliability of Mr. Banerjee's claim about repacking the boxes.

- **Sentence 2** *"Mr. Banerjee admitted that he had opened only 5 or 6 . . ."*
  Directly challenges Mr. Banerjee's report and supports Mr. Gupta's statement regarding the condition of the boxes.

- **Sentence 3** *"Both parties knew that they were talking about the same 20 boxes . . ."*
  Reinforces the argument that Mr. Banerjee's report was inaccurate.

**Sentence Contradicting the Decision in Red**

- **Sentence 4** *The learned Solicitor-General, however, attempted to argue . . ."*
  Suggests a possible gap in evidence regarding whether the boxes Mr. Gupta examined were indeed the same as those reported on by Mr. Banerjee.

- **Sentence 5** *"It was also suggested on behalf of the respondents that . . ."*
  Implies that Mr. Gupta might not have acknowledged all relevant correspondence from Mr. Banerjee.

## 7  Conclusion

We propose NCwR-Cost and NCwR-Cov, novel GNN architectures for integrating reject option in node classification. These models can reject from making predictions whenever they are not certain about predicting an example. Our experimental results show that the proposed models perform better than the baseline methods. Such results show the importance of separate GNN architecture having a reject option integrated into it. Our results on the LJP task show that these models are very effective in such applications.

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

## A    Applications on medical domain

We added applications on two more tabular health datasets: UCI Thyroid Quinlan (1986) dataset and Pima Indians Diabetes dataset Smith et al. (1988). Each tabular dataset was transformed into a graph structure to enable graph-based learning. Initially, the dataset was standardized to normalize the feature values across samples, facilitating the $k$-nearest neighbors (KNN) process. Using KNN, an edge was created between each data point and its closest neighbors in feature space which formed a sparse graph. This approach effectively connected each node (data point) with a fixed number of neighbors (taken $k = 5$), creating edges that reflect the similarity in feature space.

To represent this as a graph, each data point was treated as a node, where its feature vector constituted node attributes, and the class label served as the target variable. Edges were defined based on KNN relationships, where nodes shared edges if they were among each other's nearest neighbors. This graph was then formatted into a structure suitable for graph neural networks, containing nodes, edges and labels to support message-passing and learning across connected nodes.

The graph was constructed by concatenating the training (85%) and test (15%) sets, treating them as a unified structure to facilitate transductive learning. Approximately 10% of the training data was reserved as a validation set, ensuring a fair evaluation of the model's performance during training. While the full graph was available for message passing during training and testing, only the labels of the training nodes were utilized for model optimization. We have shown the results for NCwR-Cost and NCwR-Cov models on the UCI Thyroid dataset and Pima Indians Diabetes dataset in Table 4.

| | | **NCwR-Cost** | | **NCwR-Cov** | |
|---|---|---|---|---|---|
| **Dataset** | $d$ | Acc (%) | Cov (%) | Acc (%) | Cov (%) |
| | 0.05 | $96.56 \pm 0.68$ | $60.52 \pm 9.21$ | $99.66 \pm 0.09$ | 0.5 |
| | 0.10 | $96.40 \pm 0.13$ | $82.37 \pm 2.39$ | $99.53 \pm 0.06$ | 0.6 |
| UCI Thyroid | 0.20 | $95.29 \pm 0.40$ | $94.09 \pm 0.84$ | $99.31 \pm 0.10$ | 0.7 |
| | 0.30 | $94.37 \pm 0.21$ | $98.20 \pm 0.18$ | $98.75 \pm 0.09$ | 0.8 |
| | 0.40 | $93.75 \pm 0.18$ | $99.76 \pm 0.13$ | $97.48 \pm 0.04$ | 0.9 |
| | 0.30 | $88.54 \pm 0.68$ | $64.00 \pm 2.57$ | $94.38 \pm 1.40$ | 0.5 |
| | 0.35 | $87.40 \pm 0.13$ | $73.23 \pm 2.79$ | $90.77 \pm 1.40$ | 0.6 |
| Pima Indians Diabetes | 0.40 | $86.46 \pm 0.40$ | $84.00 \pm 3.19$ | $88.89 \pm 1.57$ | 0.7 |
| | 0.45 | $84.11 \pm 0.21$ | $92.92 \pm 1.75$ | $87.31 \pm 1.72$ | 0.8 |
| | 0.50 | $81.79 \pm 0.18$ | $99.69 \pm 0.69$ | $85.86 \pm 2.25$ | 0.9 |

Table 4: Accuracy and coverage for NCwR-Cost (varying cost of rejection $d$) and NCwR-Cov (varying desired coverage) on two health datasets.

# B  Comparison with different base GNN

We compare the results of our method on various GNN Architectures and Hyperparameters. We show that our method is GNN agnostic and results in similar performance for most of these architectures. Results are presented in Table 5.

| Coverage | GCN | GAT | GraphSAGE | GATv2 | GAT (3 layers) | GAT (4 layers) |
|---|---|---|---|---|---|---|
| 0.7 | 88.86 | 88.71 | 85.86 | 89.14 | 88.75 | 89.21 |
| 0.75 | 88.00 | 87.73 | 85.47 | 88.60 | 87.35 | 88.25 |
| 0.8 | 85.62 | 86.87 | 84.50 | 86.37 | 87.75 | 86.18 |
| 0.85 | 85.53 | 85.06 | 83.06 | 85.65 | 85.96 | 85.81 |
| 0.9 | 83.89 | 84.67 | 82.56 | 84.22 | 84.45 | 84.03 |

Table 5: Accuracy comparison of our method with various base GNN Architectures. Unless mentioned, we use two GNN layers in each architecture.

# C  Sensitivity Analysis on $\lambda$ (in NCwR-Cov)

We analyze the effect of $\lambda$ parameter on the coverage rates of the model. This constraint will penalize the model during training whenever the coverage is below the targeted. We present the results of our model without post-training calibration to measure the impact of $\lambda$ with a fixed $\tau = 0.5$ and desired coverage is 0.8. Ideally, the model coverage should be $> 80\%$ while maintaining the least difference.

| $\lambda$ | Accuracy (%) | Coverage (%) | Coverage Gap (%) |
|---|---|---|---|
| 4 | 92.42 | 66.34 | $-13.66$ |
| 8 | 91.19 | 73.25 | $-6.75$ |
| 16 | 89.81 | 79.11 | $-0.89$ |
| 32 | 88.78 | 82.42 | $+2.42$ |
| 64 | 88.13 | 82.13 | $+2.13$ |

Table 6: Effect of $\lambda$ on model coverage without post training calibration (target coverage = 80 %).

