# OpenReview forum: "Node Classification With Reject Option"
_TMLR — Accepted by TMLR_

### Review · Reviewer_b3Eq · 2025-01-27

**Summary Of Contributions:**

The authors proposed two approaches for node classification with a reject option, tailored specifically for graph attention networks (but can also be implemented in other GNN models). The coverage-based approach uses coverage as input to determine the optimal model for a specified coverage rate, while the cost-based approach identifies the best classifier based on a given rejection cost. The authors then evaluated the performance of these methods across multiple datasets. The authors also discuss an application in legal judgement prediction cases.

**Audience:**

Yes

**Broader Impact Concerns:**

No concern.

**Claims And Evidence:**

No

**Requested Changes:**

Please address the weaknesses I mentioned before.

**Strengths And Weaknesses:**

**Strengths:**

1. The paper is straightforward and easy to follow.
2. An interesting application of classification with a reject option on GNN models.
3. The authors provide long detailed discussions on the experiment results.
4. A use case of the model experiments on legal judgement prediction is provided.

**Weaknesses:**

1. The methods presented in the paper are rather direct applications of previous research in the graph ML settings. The coverage model is based on SelectiveNet (Geifman & El-Yaniv, 2019) whereas the cost-based model is based on (Cao et al., 2022).
2. The authors did not provide a clear explanation on why graph ML settings are different from regular ML settings in which the previous research was applied to. What are the challenges and difficulty of applying non-graph models on classification with a reject option to graph use cases; and how do the authors address those challenges. These details are not in the paper, which makes the contribution of the paper look limited.
3. The cost-based method is shown to have better performance in the experiments, even in the settings of optimizing for coverage. Could the author explain this further?
4. The explainability results in section 6.2 are a bit confusing. The proposed models do not directly use the text data in the model. However, the explainability results provided analyze the explainability from the part of text perspective. No internal of the model seems to be utilized in producing the explainability results. Could the authors clarify this?
5. Figure 3 is hard to see as it is too small.

---

> ### Author Response · Authors · 2025-02-28
>
> Dear Reviewer b3Eq,
>
> We appreciate your response and detailed analysis of our work. Here is our response to some of the weaknesses mentioned.
>
> W1 and W2: The methods we presented are a direct extension of existing methods for Graph Neural Networks. However, we would like to emphasize the effectiveness of GNNs in some cases instead of traditional reject option classifiers where a relation between nodes is present. For example, we can see the improvements of using NCwR-Cov over SelectiveNet but converting the Pima Diabetes dataset into a graph as mentioned in the paper.
>
> | Coverage | SelectiveNet (MLP) | NCwR-Cov |
> |----------|--------------------|----------|
> | 0.9      | 85.25%             | 85.86%   |
> | 0.8      | 86.42%             | 87.31%   |
> | 0.7      | 87.35%             | 88.89%   |
> | 0.6      | 89.11%             | 90.77%   |
>
>
> W3: The objective function of coverage based model ensures that the coverage of the model is at-least a user defined coverage value. This constraint will penalize the model during training whenever the coverage is below what we wanted. After training, we do a post-training calibration of $\tau$ to match the coverage of the test data to our expected coverage. Due to this, the coverage of the test data does not depend on the constraint $\lambda$ used during training.  This could be one reason the cost-based model is performing better than the coverage-based model consistently.
>
> W4: We regret not mentioning this clearly in the section. The explainability results using SHAP are extracted from a BERT classifier trained on this dataset. We wanted to present that when our reject option classifier rejects a sample, there are features in the input leaning toward both decisions. While handling such cases with ambiguity, unlike a traditional classifier, our approach will have a better understanding of when to make a prediction and when to abstain.
>
> W5: We have corrected this issue in the new version.
>
> Our apologies for the late response. We are open to making any more changes if required.
>
> Sincerely,
> Authors of Paper3854

---

### Review · Reviewer_7ojr · 2025-02-03

**Summary Of Contributions:**

This paper introduces NCwR, a novel framework that integrates a reject option into Graph Neural Networks (GNNs) for node classification. The method is developed with two variants: (i) NCwR-Cov, a coverage-based, and (ii) NCwR-Cost, a cost-based model. The authors evaluate their approach on Cora, Citeseer, and Pubmed citation networks, and the Indian Legal Documents Corpus (ILDC) dataset.

**Audience:**

Yes

**Claims And Evidence:**

Yes

**Requested Changes:**

Overall, I believe this paper addresses an important problem and could be a valuable contribution to the community. However, in its current form, I am inclined to reject it.

To strengthen the manuscript, I recommend that the authors:
- Improve readability, particularly in sections with dense mathematical notation, to enhance clarity and accessibility.
- Provide a more detailed discussion on the impact of coverage constraints and rejection cost, including a sensitivity analysis to illustrate how these parameters affect model performance.


Addressing these issues would significantly improve the paper’s clarity and robustness, making it more suitable for publication.

**Strengths And Weaknesses:**

Strengths:
- The "rejection option" is an intresting and fundamental task in several high-risk application, and it is not well explored in GNNs
- The study benchmarks the two proposed methods against baselines, showing competitive performance. The experiments on real-world legal data (ILDC) add practical significance.

Weaknesses
- I found the writing of the article a bit difficult to follow, several concepts are repeated, and I found it unusual to see some fine grain detains the main (i.e. sections 4.3 and 4.4)
- The impact of λ (coverage constraint) and d (rejection cost) on final accuracy and rejection behavior is not thoroughly explored. A sensitivity analysis would strengthen the claims regarding robustness

---

> ### Author Response · Authors · 2025-02-28
>
> Dear Reviewer 7ojr,
>
> $\textbf{Improving Readability}$: We appreciate your response and detailed analysis of our work. We put a lot of work into improving the overall readability and removing unnecessary details to make it a lot more efficient. We updated the manuscript with a new version.
>
> $\textbf{Sensitivity Analysis}$: We want to discuss the impact of the coverage constraint $\lambda$. This constraint will penalize the model during training whenever the coverage is below what we wanted. After training, we do a post-training calibration of \tau to match the coverage of the test data to our expected coverage. Due to this, the coverage of the test data does not depend on the constraint $\lambda$ used during training. However, we also present the results of our model without post-training calibration to measure the impact of $\lambda$ with a fixed $\tau=0.5$ and coverage=0.8.
>
> | $\lambda$ | Accuracy | Coverage |
> |---------|----------|----------|
> | 2       | 92.42%   | 66.34%   |
> | 8       | 91.19%   | 73.25%   |
> | 16      | 89.81%   | 79.11%   |
> | 32      | 88.78%   | 82.42%   |
> | 64      | 88.13%   | 82.13    |
>
>
> We presented the impact of the cost of rejection $d$ on performance in Table 2. Let us know if you meant anything different.
>
> Our apologies for the late response. We are open to making any more changes if required.
>
> Sincerely,
> Authors of Paper3854

---

### Review · Reviewer_AxiR · 2025-02-14

**Summary Of Contributions:**

This paper studies transductive node classification on graphs using GNNs under reject option settings, i.e. when the model is allowed to avoid making predictions if the uncertainty is high. The authors consider two major approaches, cost-based and coverage-based, which are established standards for reject option classifiers. To apply these two approaches for transductive classification, the authors take a GNN to produce node embeddings and then apply objective function of [1] for coverage-based approach and of [2] for cost-based approach, respectively, on top of the embeddings. The authors show experiments on Cora, Citeseer, and Pubmed datasets with standard splits, and argue that the presented methods outperform a simple baseline that rejects if the predicted class scores are lower than a threshold hyperparameter and a conformal prediction-based baseline that rejects if more than one labels are predicted. The authors also analyze the behaviors of the two methods depending on coverage rates and rejection costs, and also show by visualization that the rejected examples tend to lie around class boundaries. The authors demonstrate the two methods on legal judgement prediction task which is framed as transductive classification, and provides an explanability result based on SHAP.

[1] Geifman and El-Yaniv, SelectiveNet: A Deep Neural Network with an Integrated Reject Option (2019)

[2] Cao et al. Generalizing Consistent Multi-Class Classification with Rejection to be Compatible with Arbitrary Losses (2022)

**Audience:**

Yes

**Broader Impact Concerns:**

I have no particular broader impact concerns.

**Claims And Evidence:**

Yes

**Requested Changes:**

I would like to propose adjusting the submission to address W1-W6.

**Strengths And Weaknesses:**

Strengths

- S1. This paper studies a practically relevant problem of transductive classification under rejection option.

Weaknesses

- W1. The technical contributions are weak, as the proposed methods are direct applications of the objective functions developed in prior work [1, 2] on node embeddings from a GNN. It is not clear how the presented methods technically differ from or advance these prior work.
- W2. It is unclear why the objective functions from [1, 2] are particularly well-suited for GNN integration over other existing methods including the ones cited in the paper.
- W3. Several standard baselines such as Monte Carlo dropout (used in [1]) are missing.
- W4. The chosen datasets (Cora, Citeseer, and Pubmed) are small-scale citation datasets with very high structural homophily (e.g. see Table 5 of [3]). Thus the presented evidences are not strong enough to claim that, for example, NCwR-Cost is a superior model compared to baselines (Section 5.1) for transductive classification in general.
- W5. Experimental observations in Section 5 contribute limited knowledge, e.g. Section 5.2 mainly discusses cost/coverage-accuracy tradeoff which is not particularly unexpected, and Section 5.3 closely follows the t-SNE visualization of [1] in methodology and observations.
- W6. It is not clear how SHAP is exactly being used to produce explanations of the model predictions given in Figures 5-6. As far as I understand the model is a GAT on case-level embeddings produced by a pretrained XLNet, and I am not sure how token-level scores as in Figures 5-6 can be obtained from this type of model architecture.

[1] Geifman and El-Yaniv, SelectiveNet: A Deep Neural Network with an Integrated Reject Option (2019)

[2] Cao et al. Generalizing Consistent Multi-Class Classification with Rejection to be Compatible with Arbitrary Losses (2022)

[3] Platonov, Characterizing Graph Datasets for Node Classification: Homophily-Heterophily Dichotomy and Beyond (2022)

---

> ### Author Response · Authors · 2025-02-28
>
> Dear reviewer AxiR,
>
> We appreciate your review and the detailed drawbacks of our work. We would like to respond to the mentioned weakness below
>
> W1: While it is true that our work leverages objective functions introduced in [1, 2], our main contribution is to integrate these objectives in a GNN framework.
>
> W2: We presented two approaches for integrating a reject option in GNNs. We proposed coverage-based and cost-based models with strong approaches in existing literature of ROC. We also emphasize the effectiveness of GNNs in some cases instead of traditional reject option classifiers where a relation between nodes is present. For example, we can see the improvements of using NCwR-Cov over SelectiveNet but converting the Pima Diabetes dataset into a graph as mentioned in the paper.
>
> | Coverage | SelectiveNet (MLP) | NCwR-Cov |
> |----------|--------------------|----------|
> | 0.9      | 85.25%             | 85.86%   |
> | 0.8      | 86.42%             | 87.31%   |
> | 0.7      | 87.35%             | 88.89%   |
> | 0.6      | 89.11%             | 90.77%   |
>
>
> W3: We appreciate the reviewer’s suggestion regarding standard baselines. In response, we have incorporated experiments with Monte Carlo dropout (using a dropout rate of p = 0.5 across 10 iterations) and present our comparison on the Citeseer Dataset.
>
> | Coverage | Monte Carlo dropout | NCwR-Cov |
> |----------|---------------------|----------|
> | 0.9      | 71.16%               | 72%       |
> | 0.8      | 71.81%              | 72.8%     |
> | 0.7      | 73.52%                | 75.9%     |
> | 0.6      | 77.3%%                  | 79.6%      |
>
>
> W4: We acknowledge that datasets such as Cora, Citeseer, and Pubmed are limited by their small scale. We also experimented with ILDC, which is a large, complex real-world dataset. Results on this show that our method is not only effective on standard benchmarks but is also ready for deployment in more diverse and challenging settings.
>
> W5: We discussed all the details we felt were necessary for the methods we are presenting. We are open to adding any more specific details as required.
>
> W6: W4: We regret not mentioning this clearly in the section. The explainability results using SHAP are extracted from a BERT classifier trained on this dataset. We wanted to present that when our reject option classifier rejects a sample, there are features in the input leaning toward both decisions. While handling such cases with ambiguity, unlike a traditional classifier, our approach will have a better understanding of when to make a prediction and when to abstain.
>
> Our apologies for the late response. We are open to making any more changes if required.
>
> Sincerely,
> Authors of Paper3854

---

### Decision · Action_Editor_te6m · 2025-04-13

**Recommendation:** Accept with minor revision

**Comment:**

This paper does meet the bar for TMLR acceptance since it validates the claim on small-scale graphs.

**Audience:**

Some portion of the researchers using GNNs for transductive node classification would be interested in the paper.

The audience can be broadened by clearly demonstrating the practical relevance of the work.

**Claims And Evidence:**

The authors claim that the existing ideas of [1] and [2] works well for transductive node classification tasks. The claims are validated for small-scale graphs (Citeseer/Cora/PubMed/ILDC with 7593 nodes).

[1] Geifman and El-Yaniv, SelectiveNet: A Deep Neural Network with an Integrated Reject Option (2019)
[2] Cao et al. Generalizing Consistent Multi-Class Classification with Rejection to be Compatible with Arbitrary Losses (2022)

As the reviewers mentioned in the final decision phase, I recommend the authors to conduct large-scale experiments (AxiR) and sensitivity analysis on rejection cost (b3Eq) in the revised manuscript.

---

> ### Author Response · Authors · 2025-05-15
> **Camera Ready Submission**
>
> Dear Sungsoo Ahn,
>
> We performed experiments on both our methods on ogbn-arxiv dataset and added the results. The sensitivity analysis on rejection cost is present in the previous version, we also added a sensitivity analysis of $\lambda$ on coverage. We submitted the revised camera ready version. We appreciate your inputs to improve our paper.
>
> Sincerely, Authors of Paper3854